# Evaluation of an Active Disturbance Rejection Controller for Ophthalmic Robots with Piezo-Driven Injector

**DOI:** 10.3390/mi15070833

**Published:** 2024-06-27

**Authors:** Qiannan Tao, Jianjun Liu, Yu Zheng, Yang Yang, Chuang Lin, Chenhan Guang

**Affiliations:** 1School of Energy and Power Engineering, Beihang University, Beijing 100191, China; tqn_lea@163.com; 2School of Mechanical Engineering and Automation, Beihang University, Beijing 100191, China; yyds_l@vip.163.com (J.L.); drlin@leadmicros.com (C.L.); 3College of Automation and College of Artificial Intelligence, Nanjing University of Posts and Telecommunications, Nanjing 210023, China; 4School of Mechanical and Materials Engineering, North China University of Technology, Beijing 100144, China; guangchenhan@ncut.edu.cn

**Keywords:** retinal vein cannulation, piezo-driven injector, ophthalmic robot, active disturbance rejection controller

## Abstract

Retinal vein cannulation involves puncturing an occluded vessel on the micron scale. Even single millinewton force can cause permanent damage. An ophthalmic robot with a piezo-driven injector is precise enough to perform this delicate procedure, but the uncertain viscoelastic characteristics of the vessel make it difficult to achieve the desired contact force without harming the retina. The paper utilizes a viscoelastic contact model to explain the mechanical characteristics of retinal blood vessels to address this issue. The uncertainty in the viscoelastic properties is considered an internal disturbance of the contact model, and an active disturbance rejection controller is then proposed to precisely control the contact force. The experimental results show that this method can precisely adjust the contact force at the millinewton level even when the viscoelastic parameters vary significantly (up to 403.8%). The root mean square (RMS) and maximum value of steady-state error are 0.32 mN and 0.41 mN. The response time is below 2.51 s with no obvious overshoot.

## 1. Introduction

Retinal vein occlusion (RVO) occurs when a blockage of one or more small veins carries blood away from the retina. RVO has been reported to be the second most common retinal vascular occlusive disease [1]. Its impact on vision can be severe, causing blurred and distorted vision [2]. Currently, there is no available treatment to resolve RVO directly. A possible treatment is retinal vein cannulation (RVC), which involves the insertion of a cannulation needle and light pipe through the scleral ports (diameter: about 0.6 mm) to inject a clot-dissolving drug into the retinal vein. Figure 1 gives a visual representation of the procedure of RVC.

Performing micro-surgical tasks in RVC requires highly specialized skills that are at the limits of human motor capabilities. The surgeons’ hand tremor amplitudes exceeds 150 μm [3], which is significantly larger than the diameters of the retinal veins (50–150 μm [4]). The surgeons are not able to perceive the contact force at the needle tip, which is below 10 mN [5]. The limited motor and force control, degraded manual dexterity, and possible visual, physical, and mental fatigue of a surgeon may intensify their hand tremors and increase the risk of sight-damaging iatrogenic trauma to the eye.

A technically feasible remedy to these challenges is robot-assisted RVC. Figure 1 illustrates the three processes involved in robot-assisted RVC: (1) Approach—where the robot drives the needle to approach the retinal vein. (2) Contact—where the robot ensures that the contact force reaches the expected value, making a puncture into the vein without causing any damage to the retina. (3) Inject—where the robot keeps the needle steady and injects the clot-dissolving drug into the retina vein.

Current robots designed for robotic-assisted vascular catheterization (RVC) have primarily focused on improving the precision of surgical tools and filtering out the surgeon’s hand tremors. For instance, leveraging the natural steady motion of robotic arms [6,7,8], compensating the tool tip’s motion with hand-held devices [9], wireless magnetic control of an untethered microrobot [10], localization of the tooltip with OCT [11]/computer vision [12], and sensing tool–tissue proximity with bio-impedance [13]. While precise needle movement is the foundation of robot-assisted RVC, the lack of a contact-force controller presents a safety challenge. A robotic system which is capable of sensing and controlling the contact force is advantageous.

Researchers from Johns Hopkins University have developed force-sensing tools that utilize fiber Bragg grating (FBG) sensors. With these sensors, the tools can measure the contact force, scleral force, and insertion depth. The force measurement can be integrated into the robot control law [14,15] or provide auditory substitution [16]. Dual force constraint controllers [17], and hybrid position/force controllers [18] have also been proposed to keep the force within desired range. In the contact process, the robot is expected to track the desired force during the contact process, not just limit the contact force.

Ebrahimi et al. [19] model the sclera with a linear elastic contact model and propose an adaptive controller to track the sclera force. However, the stiffness fails to describe the retina vein’s viscoelastic characteristics. Additionally, the viscoelastic parameters of eye tissue are typically related to a variety of factors, such as age, gender, and species (human or different experimental animals) [20], making the contact force highly uncertain and complex. To improve the force control performance, a viscoelastic contact model and a robust controller are necessary.

The parameter uncertainty could be regarded as an inner disturbance in the viscoelastic contact model. Active disturbance rejection control, presented by Han [21], is a control strategy with strong adaptability, robustness, and independence from the accurate parameters of the system model. Active disturbance rejection control is proven to have good rejection ability against disturbance and noise [22,23]. To the best of our knowledge, active disturbance rejection control has not been applied to reject the inner disturbance in the viscoelastic contact model and control the “needle–retina vein” contact force.

This preliminary study establishes a contact model based on viscoelastic theory to capture the nonlinear, uncertain behavior of vessels. Then, based on the viscoelastic contact model, an active disturbance rejection force controller (ADRC) is designed to reject the disturbance caused by the parameter uncertainty, and precisely control the contact force. Section 2 introduces the proposed ophthalmic surgical robot and builds the viscoelastic contact model. Section 3 focuses on the design of the ADRC. Section 4 analyzes the disturbance rejection ability through simulation, while Section 5 presents the experimental setup and results. Section 5 concludes this paper.

## 2. Materials and Methods

### 2.1. Ophthalmic Surgical Robot

Figure 2a,b give the components and mechanism configuration of the ophthalmic surgical robot system, respectively. The proposed robot can control the instrument’s orientation, insertion, and rotation. The instrument’s orientation is adjusted by four linear motors (LM1-4, MSR080, Paker, CO, USA). We can rotate the instrument around or along axis *y*_B_ by changing the stroke of LM1 and LM3. Similarly, we can rotate the instrument around or along axis *x*_B_ by changing the stroke of LM2 and LM4. The proposed ophthalmic surgical robot can also perform continuous curvilinear capsulorhexis or membrane peeling with forceps [8,24].

The piezo-driven injector consists of a piezoelectric linear stage (CONEX-SAG-LS48P, NewPort, RI, USA, P_5_), a rotation motor (DCX10S, Maxson, Sachseln, Switzerland, R_5_), and the instrument shown in Figure 2c. The positioning accuracy of the linear motors and piezoelectric linear stage are 0.01 mm and 25 nm, respectively. The detailed components of the instrument are given in Figure 2c. The instrument consists of a force sensor, a connector, a Luer fitting, and a cannulation needle (41G). The needle is also equipped with three FBG sensors. The force sensor (LSB200-20g, Futek, CA, USA, precision: 0.2 mN) aims to measure the axial force (*z*, Figure 2b,c) acting at the needle.

### 2.2. Viscoelastic Contact Model

Figure 3 illustrates the viscoelastic contact model, which models the contact force *f*(*t*) with two parallel Maxwell viscoelastic models [25]. Each Maxwell model combines a spring with a damper in serial.

The mathematical description of the viscoelastic contact model is given in Equation (1).
(1)$$f\left(t\right)=\left(E_{1}+E_{2}\right)\frac{dx\left(t\right)}{dt}-\left(\tau_{1}+\tau_{2}\right)\frac{df\left(t\right)}{dt}$$
where *x*(*t*) is the axial movement of PLS1, *τ_i_* = *η_i_*/*E_i_*, *i* = 1, 2. *E* and *η* represents the spring stiffness constant and the damping factor, respectively.

*E* and *τ* are the mechanical parameters of the retinal vein and can be written as below:$$E=E_{ave}+\Delta E,\tau =\tau_{ave}+\Delta \tau ,\;E=E_{ave}+\Delta E,\;\tau =\tau_{ave}+\Delta \tau$$
where ave means the mean value measured by experiments, and Δ*E* and Δ*τ* are the uncertainty caused by many factors, such as age, gender, experiment models (pig/rabbit/human, etc.).

Equation (1) can be rewritten in the frequency domain:(2)$$F\left(s\right)=\left(\frac{E_{1}\tau_{1}s}{1+\tau_{1}s}+\frac{E_{2}\tau_{2}s}{1+\tau_{2}s}\right)X\left(s\right)$$
where *s* is the Laplace operator.

The step response of Equation (2) in the time domain is shown below:(3)$$f\left(t\right)=E_{1}e^{-t/\tau_{1}}+E_{2}e^{-t/\tau_{2}}$$

Equation (3) is used to identify *E_i_*_,ave_, *τ_i_*_,ave_, *i* = 1, 2 in the contact model, which is given in Section 5.

Δ*E* and Δ*η* are the contact model’s main disturbances, making it hard to control *f*(*t*) precisely. In this paper, an LADRC-based force controller with disturbance-rejection ability is first presented and designed for a viscoelastic contact model.

### 2.3. Active Disturbance Rejection Force Controller

Figure 4 gives the block diagram of the ADRC. The ADRC consists of 3 parts: linear extended state observer (LESO), linear state error feedback (LSEF), and a numerical calculation module based on the fourth-order Runge–Kutta (RK4) method. This part gives the detailed design of the controller.

First, we rewrite Equation (2) into Equation (4):(4)$$\frac{F\left(s\right)}{X\left(s\right)}=\frac{b_{1}s}{1+a_{1}s}+\frac{b_{2}s}{1+a_{2}s}$$
where $\eta_{i}=E_{i}\tau_{i}$, $a_{i}=\tau_{i},\;b_{i}=\eta_{i},$
*i* = 1, 2

Then, we can obtain Equation (5) by dividing *s* on both sides of Equation (4).
(5)$$\frac{Y\left(s\right)}{U\left(s\right)}=\frac{F\left(s\right)}{v\left(s\right)}=\frac{b_{1}}{1+a_{1}s}+\frac{b_{2}}{1+a_{2}s}$$
where *Y*(*s*) is regarded as the output of the viscoelastic contact model. *v*(*s*) is the velocity of the PLS1, and it is defined as the input *U*(*s*).

Equation (5) can be rewritten as below:(6)$$\frac{Y\left(s\right)}{U\left(s\right)}=\frac{\left(b_{1}+b_{2}\right)+\left(a_{1}b_{2}+a_{2}b_{1}\right)s}{1+\left(a_{1}+a_{2}\right)s+a_{1}a_{2}s^{2}}=\frac{B_{1}s+B_{2}}{s^{2}+A_{2}s+A_{3}}$$

Next, we define the median parameter *Y*_1_(*s*) in Equation (7):(7)$$Y_{1}\left(s\right)=\frac{1}{s^{2}+A_{2}s+A_{3}}U\left(s\right)$$

Submitting Equation (7) into Equation (6), we obtain Equation (8):(8)$$Y\left(s\right)=Y_{1}\left(s\right)\left(B_{1}s+B_{2}\right)$$

We obtain Equations (9) and (10) by rewriting Equations (7) and (8) in the time domain. Equation (9) is the state equation, and Equation (10) is the observation equation.
(9)$${\ddot{y}}_{1}=u-A_{2}{\dot{y}}_{1}-A_{3}y_{1}$$
$$y=B_{1}{\dot{y}}_{1}+B_{2}y_{1}$$

Uncertainty (Δ*E* and Δ*τ*) exists in both the state equation and observation equation. In this paper, we only deal with the uncertainty in the state equation. Then, we obtain Equation (11) by representing the uncertainty (Δ*E* and Δ*τ*) with *ω.*
(10)$$\begin{array}{l} {\ddot{y}}_{1}=f+b_{0}u\\ y=B_{1}{\dot{y}}_{1}+B_{2}y_{1}\\ f=-A_{2}{\dot{y}}_{1}-A_{3}y_{1}+\omega +\left(1-b_{0}\right)u \end{array}$$
where *b*_0_ is the known part in *u*, *f* is the sum of the uncertainty and unknown disturbance.

Next, we define the state variable $x={\left[\begin{array}{ccc} y_{1} & {\dot{y}}_{1} & f \end{array}\right]}^{T}$ and rewrite Equation (11) as Equation (12).
(11)$$\begin{array}{l} x=Ax+Bu+E\dot{f}\\ y=Cu \end{array}$$
where
$$A=\left[\begin{array}{ccc} 0 & 1 & 0\\ 0 & 0 & 1\\ 0 & 0 & 0 \end{array}\right],B=\left[\begin{array}{c} 0\\ b_{0}\\ 0 \end{array}\right],E=\left[\begin{array}{c} 0\\ 0\\ 1 \end{array}\right],C=\left[\begin{array}{ccc} B_{2} & B_{1} & 0 \end{array}\right]$$

Then, the LESO can be obtained as below:(12)$$\begin{array}{l} \dot{z}=Az+Bu+L\left(y-Cz\right)\\ \hat{y}=Cz \end{array}$$
where ***z*** is the state variable in LESO and aims to observe ***x***.

Then, we rewrite Equation (13) as Equation (14) by ignoring the differentiation of *f*, since it can be regarded as a part of the disturbance.
(13)$$\begin{array}{l} \dot{z}=\left(A-LC\right)z+\left[B,L\right]u_{c}\\ y_{c}=z \end{array}$$
where $u_{c}={\left[\begin{array}{cc} u & y \end{array}\right]}^{T}$, ***y****_c_* is the output of the LESO.

In Equation (14), ***L*** is the gain matrix of the LESO, defined as below:(14)$$L={\left[\begin{array}{ccc} 3\omega_{0} & 3\omega_{0}^{2} & \omega_{0}^{3} \end{array}\right]}^{T}$$
where *ω*_0_ is the bandwidth of the LESO.

In this paper, we solve Equation (14) with RK4. Equation (15) gives the formula of the LSEF.
(15)$$\begin{array}{l} u_{0}=k_{p}\left(r-z_{1}\right)-k_{d}z_{2}\\ u=\frac{u_{0}-z_{3}}{b_{0}} \end{array}$$
where $k_{p}=\omega_{c}^{2},k_{d}=2\omega_{c}$. *z*_1_, *z*_2_, and *z*_3_ are obtained from the LESO, *ω_c_* is the bandwidth of the LSEF, *r* is the desired force, and *u* is the control value. *u* is the velocity of PLS1.

## 3. Disturbance Rejection Capability Analysis

This part analyzes the disturbance-rejection capability of the proposed controller. First, we obtain Equation (16) by expanding Equation (13).
(16)$$\left\{\begin{array}{l} {\dot{z}}_{1}=z_{2}-3\omega_{0}\left(z_{1}-y\right)\\ {\dot{z}}_{2}=z_{3}-3\omega_{0}^{2}\left(z_{1}-y\right)+b_{0}u\\ {\dot{z}}_{3}=-\omega_{0}^{3}\left(z_{1}-y\right) \end{array}\right.$$

Then, we obtain Equation (17) by rewriting Equation (16) in the frequency domain.
(17)$$\begin{array}{l} z_{1}=\frac{3\omega_{0}s^{2}+3\omega_{0}^{2}s+\omega_{0}^{3}}{{\left(s+\omega_{0}\right)}^{3}}y+\frac{b_{0}s}{{\left(s+\omega_{0}\right)}^{3}}u\\ z_{2}=\frac{\left(3\omega_{0}^{2}s+\omega_{0}^{3}\right)s}{{\left(s+\omega_{0}\right)}^{3}}y+\frac{b_{0}s\left(s+3\omega_{0}\right)}{{\left(s+\omega_{0}\right)}^{3}}u\\ z_{3}=\frac{\omega_{0}^{3}s}{{\left(s+\omega_{0}\right)}^{3}}y-\frac{b_{0}\omega_{0}^{3}}{{\left(s+\omega_{0}\right)}^{3}}u \end{array}$$

Submitting the LSEF into Equation (17), we obtain Equation (18):(18)$$U\left(s\right)=\frac{1}{b_{0}}G_{1}\left(s\right)\left[\omega_{c}^{2}R\left(s\right)-H\left(s\right)Y\left(s\right)\right]$$
where
$$\begin{array}{l} G_{1}\left(s\right)=\frac{{\left(s+\omega_{0}\right)}^{3}}{{\left(s+\omega_{0}\right)}^{3}+2\omega_{c}s^{2}+\left(\omega_{c}^{2}+6\omega_{c}\omega_{0}\right)s-\omega_{0}^{3}}\\ H\left(s\right)=\frac{\left(3\omega_{c}^{2}\omega_{0}+6\omega_{c}\omega_{0}^{2}+\omega_{0}^{3}\right)s^{2}+\omega_{c}^{2}\omega_{0}^{3}}{{\left(s+\omega_{0}\right)}^{3}}+\frac{\left(3\omega_{c}^{2}\omega_{0}^{2}+2\omega_{c}\omega_{0}^{3}\right)s}{{\left(s+\omega_{0}\right)}^{3}} \end{array}$$
$$Y\left(s\right)=\frac{1}{s^{2}}\left(B_{2}+B_{1}s\right)\left(f+b_{0}U\right)$$

Then, we obtain Equation (19) by rewriting Equation (18).
(19)$$Y\left(s\right)=D_{r}\left(s\right)R\left(s\right)+D_{f}\left(s\right)f\left(s\right)$$
where *D_r_*(*s*) is the input sensitivity function. *D_f_*(*s*) is the disturbance sensitivity function.

Equation (20) gives the formula of *D_f_*(*s*).
(20)$$D_{f}\left(s\right)=\frac{k_{4}s^{4}+k_{3}s^{3}+k_{2}s^{2}+k_{1}s}{m_{5}s^{5}+m_{4}s^{4}+m_{3}s^{3}+m_{2}s^{2}+m_{1}s+m_{0}}$$
where
$$k_{4}=B_{1},\;k_{3}=B_{2}+5B_{1}\omega_{0},\;k_{2}=4B_{1}\omega_{0}^{2}+5B_{2}\omega_{0}+6B_{1}\omega_{0}\omega_{c},\;k_{1}=B_{2}\omega_{0}\left(4\omega_{0}^{\text{​}}+6\omega_{c}\right),$$
$$m_{5}=1,\;m_{4}=2\omega_{c}+3\omega_{0},\;m_{0}=B_{2}\omega_{c}^{2}\omega_{0}^{3},\;m_{3}=3B_{1}\omega_{c}^{2}\omega_{0}+\omega_{c}^{2}+6B_{1}\omega_{0}^{2}\omega_{c}+6\omega_{c}\omega_{0}+B_{1}\omega_{0}^{3}+3\omega_{0}^{2}$$
$$m_{2}=3B_{1}\omega_{c}^{2}\omega_{0}^{2}+3B_{2}\omega_{c}^{2}\omega_{0}+2B_{1}\omega_{c}\omega_{0}^{3}+6B_{2}\omega_{c}\omega_{0}^{2}+B_{2}\omega_{0}^{3}$$
$$m_{1}=B_{1}\omega_{c}^{2}\omega_{0}^{3}+3B_{2}\omega_{c}^{2}\omega_{0}^{2}+2B_{2}\omega_{c}\omega_{0}^{3}$$

Next, we simulate the tracking performance of *D_f_*(*s*). *D_f_*(*s*) is the transfer function between disturbance *f*(*s*) and output Y(*s*). When *f*(*s*) occurs and the steady value of *Y*(*s*) is nearly zero, it means that *f*(*s*) cannot affect Y(s), and *D_f_*(*s*) can reject the disturbance.

The disturbance rejection ability simulation is performed by MATLAB. A step signal is chosen as *f*(*s*). *ω_c_* and *ω*_0_ are hyperparameters that should be tuned manually. After tuning, *ω_c_* and *ω*_0_ are set as 16 and 60, respectively.

The range of mechanical parameters is set as below:$$E_{1}\in \left[0.1,2.0\right],E_{2}\in \left[0.01,10\right],\tau_{1}\in \left[0.1,2.0\right],\tau_{2}\in \left[0.01,10\right]$$

As shown in Figure 5, no matter what the mechanical parameters are, the value of *D_f_*(*s*)*f*(*s*) converges to zero within just one second. Changes in mechanical parameters have little effect on the output Y(*s*). The simulation results are consistent with [22]. The proposed controller is highly resistant to disturbance.

## 4. Experimental Results and Discussion

### 4.1. Experimental Setup

Figure 6 gives the experimental setup. All experiments are performed with the proposed ophthalmic surgical robot (Section 2). A force sensor (LSB200-20g, Futek, CA, USA, precision: 0.2 mN) and a surgical cannulation needle (41G, Incyto, Cheonan-si, South Korea, diameter: 90 μm) are attached to the end of the proposed robot.

Experiments using the chicken chorioallantoic membrane of 13-day-fertilized chicken embryos [5,26,27] were performed to evaluate the proposed force controller. All experiments were supervised by an expert surgeon with more than 20 years clinical experience. The surgeon made sure that the cannulation needle contacted the vessel precisely.

The cannulation needle was driven by the piezoelectric linear stage of the proposed robot. Data were collected using an industrial computer (ARK3523, Advantech, Taiwan, China), which also calculated velocity commands based on the ADRC and sent them to the controller of the piezoelectric linear stage via USB communication. The needle’s moving accuracy is approximately 25 nm. The control frequency is 200 Hz.

In this article, we perform three groups of experiments to estimate the force controller. The detailed setup of each group is given below:

I. Parameter identification. This group aims to identify the vein’s mechanical parameters (*E*_ave_ and *τ*_ave_) from step response data. In this group, the pose of the robot is chosen randomly. Five embryos are used in this group. For each embryo, we adjust the robot to contact the vein, move 0.5 mm along axis *z*, keep steady, and record the contact force.

For each embryo, we choose one contact point with 10 replicates at each point. An ophthalmic doctor carefully selects the contact point. Then, the model parameters (*E_i_*_,ave_, *τ_i_*_,ave_, *i* = 1, 2) in the viscoelastic contact model can be obtained by fitting the collected force data to Equation (3).

II. Step test. Group II uses another five embryos. For each embryo, we adjust the robot to contact the vein, and move the needle along axis *z* to reach the desired force. The desired force is set as 5 mN. For each embryo, the pose of the robot is set as three conditions: *θ*_1_ = 45°; *θ*_1_ = 30°; *θ*_1_ = 0°. For each pose, we choose three contact points with five replicates at each point.

III. Precise adjustment. Another five embryos are used in this group. In this group, the pose of the robot is the same as for group II.

For each pose, we choose two contact points with five replicates at each point. In each test, we first set the desired force between 1 mN and 5 mN. Then, we increase/decrease the desired force by 1 mN or 1.5 mN every 5 s. In group II and group III, after tuning, *b*_0_, *ω_c_*, and *ω*_0_ are set as 20, 16, and 60, respectively.

### 4.2. Results and Discussion

This part gives the experimental results. Figure 7 and Table 1 show the results of group I and group II. Figure 7a–e are the step response curve of every embryo. The fitted parameters (*E_i_*_,ave_, *τ_i_*_,ave_, *i* = 1, 2) are given in Table 1. Figure 8f–j are the results of group II.

The steady-state error (SSE) and response time are also given in Table 1. As shown in Figure 7 and Table 1, for all tested embryos, the coefficient of determination *R*^2^ is over 0.97. That means all the fitted parameters are reliable. But the values of *E*_1,ave_, *τ*_1,ave_, *E*_2,ave_, *τ*_2,ave_ have variation of over 81.2%, 73.4%, 403.8%, 172.3%, respectively. The model parameters show large interindividual differences. With the proposed controller, the SSE is [0.24, 0.35] mN. The response time ranges from 1.41 s to 2.23 s. No obvious overshoot occurs. The proposed controller is capable of rejecting the disturbance caused by the uncertainty in the contact model.

As shown in Figure 8a–e, the proposed controller can adjust the contact force at 1 mN intervals. No obvious overshot occurs in group III. The root mean square (RMS) values of the SSE of group II and group III are 0.25 mN and 0.41 mN, respectively. The response times range from 1.04 s to 2.51 s, the RMS and maximum SSE are 0.32 mN and 0.47 mN, respectively. The outliers in Figure 8 mainly come from the error in the transient process.

The experimental results indicate the proposed ADRC controller can reject the disturbance and precisely control the contact force. The RMS value of the SSE is lower than the acceptable overshot in previous works (0.9 mN) [15,17], and close to the force-sensing accuracy in previous work (about 0.3 mN) [7,14,18]. In group III, the larger error associated with “decreasing the desired force” might be attributed to the hysteresis influence of the vein. The comparison of this work and the best practices in ophthalmic surgical robots are listed in Table 2. It should be noted the desired force values of SHER and Micron are 100 mN and 6 mN, respectively.

About 80% of the vein puncture force is less than 5 mN [5]. Hence, the same value is adopted as the desired force in group II. The standard deviation of the vein puncture force is 1 mN [5]. Therefore, the same value is set as the force intervals.

In this article, the contact force is measured by a commercial force sensor, which is too large to enter the scleral port. This issue could be relieved by replacing the force sensor with a customized micro-force sensor developed by FBG, which is a part of our ongoing study. The force sensor’s precision (0.2 mN) is higher than FBG force sensors (about 0.3 mN [30]). As a result, we believe the measured force data are reliable, and the proposed controller is suitable for retina vein cannulation.

It is possible to reduce the response time by increasing *ω_c_*, but this can result in severe overshoot, which increases surgical risk. Therefore, we choose to prioritize reducing the overshoot, even if it means sacrificing response time.

We did not perform dynamic tracking experiments for the following reasons. The proposed controller aims to reject disturbance and control the force during the contact process shown in Figure 1. The expected force should precisely puncture the vein without damaging the retina. In this context, the expected force is more likely to be static and specific, rather than dynamic. Therefore, we choose to perform a step response experiment and precise adjustment experiment to evaluate the disturbance rejection ability of the ADRC.

While dynamic force might have potential benefits in RVC, this paper focuses on the disturbance rejection ability of the proposed controller. The effect of dynamic force, validating this work with more samples (in vivo/ex vivo, pig/rabbit) belongs to our ongoing study.

In our study, we use two sets of embryos, which we refer to as group I and group II. The reason for this is that in experimental studies we can collect data and identify the viscoelastic contact model multiple times. However, in clinical settings, the parameters of each tested sample can vary significantly from the identified model, resulting in increased control error.

Our proposed method can accurately identify the precise value of each tested sample based on previously collected data, allowing us to control the contact force with great precision. This precise control of the contact force is the key feature of our proposed method.

The embryos in group I are used to collect data, while the embryos in group II are the samples tested in the clinical setting. By using the data collected from group I, our proposed method can accurately control the contact force in group II, even though the embryos in group II are completely different from those in group I. For this work, the ADRC is designed such that the robot can control the contact force automatically. The level of autonomy (LoA) of the proposed solution is LoA 2 [31,32], which means the “contact” sub-task in RVC is completed automatically.

In the future, a hybrid control method will be investigated for injecting clot-dissolving drugs, which will simultaneously limit the needle’s position and contact force. This could be a potential solution to free surgeons from extremely difficult injection procedures and improve overall surgical efficiency.

## 5. Conclusions

This preliminary study introduces an ADRC to reject disturbances caused by the vessel’s uncertain viscoelastic characteristics and accurately control the “needle–vessel” contact force. We first model the nonlinear, uncertain behavior of the vessel with a viscoelastic contact model. Then, an ADRC is proposed based on the viscoelastic contact model. Fertilized chicken embryos are used to identify the viscoelastic model, and estimate the active disturbance rejection control. The results show that the ADRC can precisely control the contact force when viscoelastic parameters change greatly. The contact force can be adjusted at 1 mN intervals. The RMS and maximum value of SSE are 0.32 mN and 0.41 mN, respectively. No obvious overshoot occurs during experiments. This study provides a potential way to perform technically difficult RVC procedures.

## Figures and Tables

**Figure 1 micromachines-15-00833-f001:**
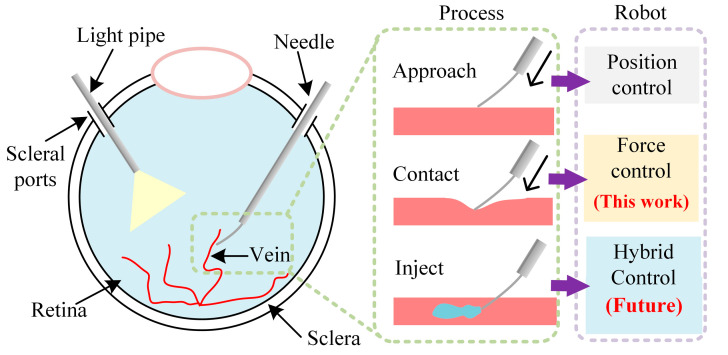
Illustration of the three processes (approach, contact, and inject) in retinal vein cannulation.

**Figure 2 micromachines-15-00833-f002:**
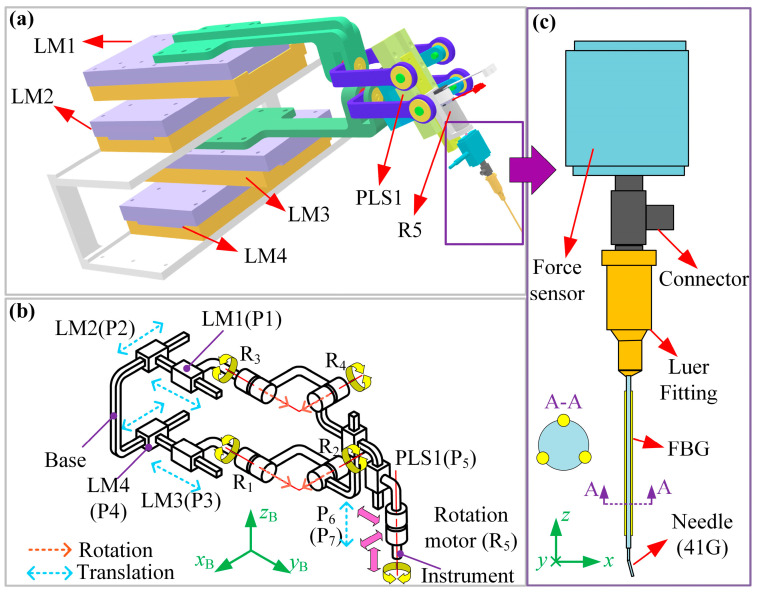
The ophthalmic robot with piezo-driven injector. (**a**,**b**) Components and configuration of the proposed ophthalmic robot. (**c**) The detailed components of the instrument.

**Figure 3 micromachines-15-00833-f003:**
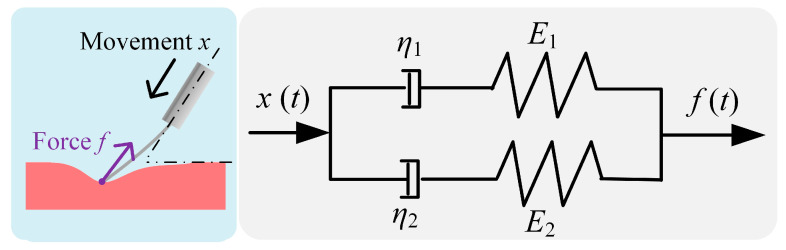
Illustration model of the viscoelastic contact model.

**Figure 4 micromachines-15-00833-f004:**
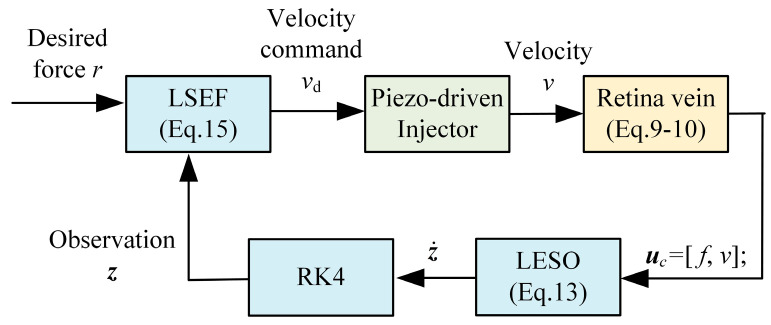
Block diagram of the ADRC.

**Figure 5 micromachines-15-00833-f005:**
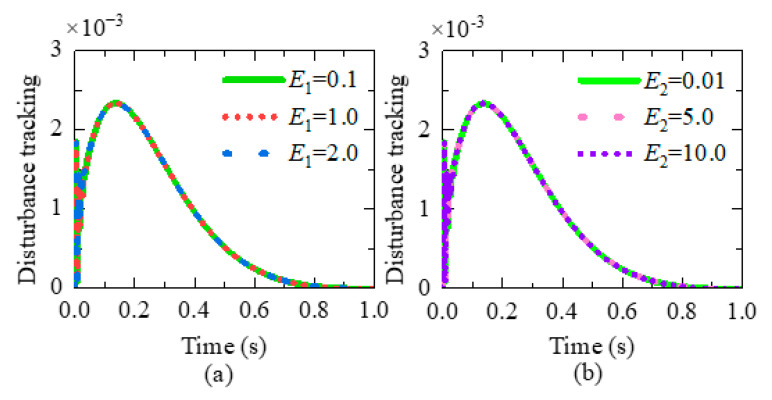
Disturbance-rejection capability of the proposed controller. (**a**) Tracking performance of *D_f_*(*s*) with different *E*_1_. (**b**) Tracking performance of *D_f_*(*s*) with different *E*_2_. (**c**) Tracking performance of *D_f_*(*s*) with different *τ*_1_. (**d**) Tracking performance of *D_f_*(*s*) with different *τ*_2_.

**Figure 6 micromachines-15-00833-f006:**
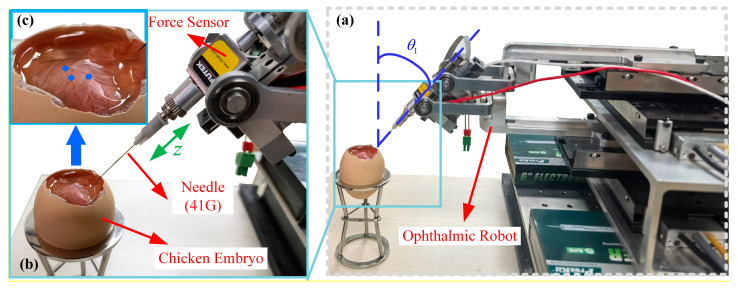
Experimental setup. (**a**) The overall view of the ophthalmic robot. (**b**) Detail description of the needle and chicken embryo. (**c**) Illustration of the contact point (blue dot) on chicken embryo.

**Figure 7 micromachines-15-00833-f007:**
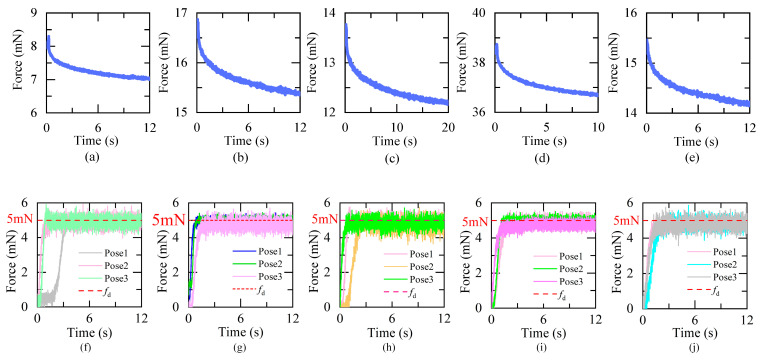
Experimental results of group I and group II. (**a**–**e**) Force release curves of embryos 1–5, respectively. (**f**–**j**) Step responses of embryos 1–5, respectively.

**Figure 8 micromachines-15-00833-f008:**
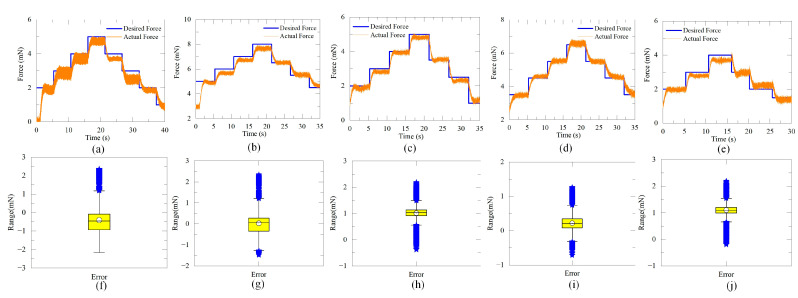
Experimental results and error of group III. (**a**–**e**) Force response of embryos 1–5, respectively. (**f**–**j**) Force error of embryos 1–5, respectively.

**Table 1 micromachines-15-00833-t001:** Experimental results of group I and group II.

	Group I					Group II	
	*E* _1_	*τ* _1_	*E* _2_	*τ* _2_	*R* ^2^	SSE (mN)	Response Time (*s*)
Embryo 1	0.692	1.042	7.405	0.004611	0.9789	0.31	1.41
Embryo 2	0.7765	1.038	15.865	0.00272	0.9834	0.29	1.93
Embryo 3	0.801	0.6156	12.665	0.002042	0.9812	0.35	1.97
Embryo 4	1.254	1.068	37.31	0.001693	0.9735	0.24	1.80
Embryo 5	0.701	0.9863	14.62	0.002621	0.9881	0.32	2.23

**Table 2 micromachines-15-00833-t002:** Comparison with other ophthalmic surgical robots.

	Tech	Max Relative Error	Surgical Procedure
JHU LCSR lab (SHER) [15]	CM	25.56%	RVC
Carnegie Mellon (Micron) [18]	HH	40%	RVC/membrane peeling
Chinese Academy of Sciences [28]	TM	O	Cataract surgery
TU Munich [6]	TM	O	RVC
Wenzhou Univ [29]	TM	O	RVC
This work	TM	9.4%	RVC

List of abbreviations: TM = telemanipulation, CM = comanipulation, HH = hand-held, O = unknown.

## Data Availability

Data is contained within the article.

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
