# Peer review of "Evaluation of an Active Disturbance Rejection Controller for Ophthalmic Robots with Piezo-Driven Injector"

_micromachines, 2024, doi:10.3390/mi15070833_

Round 1

Reviewer 1 Report

Comments and Suggestions for Authors

This paper develops an active disturbance rejection force controller for regulating the interaction force with a retinal vessel. It is a new idea in this application, and the proposed controller showed good performance. The paper overall is in good writing quality, clear, concise, and suitable for this journal. I would suggest to accept the paper. It would be great if the authors can address below minor comments:

1. A bit more explanation is suggested for Figure.5

2. It is not clear where the contact points are located in Figure 7. Please consider adding illustrations to clarify this.

4. Suggests the authors add a short explanation of the outliers in Figure 9.

Reviewer 2 Report

Comments and Suggestions for Authors

This paper addresses the critical challenge of controlling contact force in retinal vein cannulation using a robotic piezo-driven injector. Although the theory and the proposed controller generally present no novelty, the proposed viscoelastic contact model and active disturbance rejection controller are well-validated experimentally. Authors are encouraged to put their proposed solution on the Level of Autonomy scale (presented e.g., in DOI: 10.1126/scirobotics.aar7650 or  10.1109/TMRB.2019.2913282). The only deficiency of the paper is that it lacks a comparison with other state-of-the-art solutions or current best practices in the field, while there are numerous (See e.g., doi: 10.1109/JPROC.2022.3180350). Including such a comparison would provide better context for the proposed controller's performance and novelty.

The minor remarks are the following:
- Longer, more detailed explanations in figure captions would enhance their clarity
- Please annotate the same joints in Figures 2a and 2c for consistency.
- Enlarge Figure 5 for better readability.
-  Remove redundant subfigures in Figure 7, as they do not add new information beyond Figure 6.
Overall, this paper presents a promising solution with strong experimental support, but would benefit from comparative analysis and minor improvements in figure presentation.

Comments on the Quality of English Language

A general proof reading would be appropriate.

Reviewer 3 Report

Comments and Suggestions for Authors

The paper utilizes a viscoelastic contact model to explain the mechanical characteristics of retinal blood vessels and aims to solve the problem of retinal vein damage during the puncturing process. The topic is quite interesting and practically significant. However, there are still some concerns that need to be addressed:

  1. Use of Chicken Embryos:

    • The authors used chicken embryos to identify the viscoelastic model and estimate the ADRC. Is there any specific reason for applying chicken embryos? It is recommended to provide a comparison of parameters between retinal blood vessels and chicken embryos in a table, including elastic modulus, viscosity, stress-strain relationship, and creep & relaxation behavior. This will help readers understand the experiment's relevance and ensure that the experimental materials were not chosen arbitrarily.
  2. Variability of Chicken Embryos:

    • It is challenging to guarantee that each chicken embryo has the same characteristics, which might introduce bias into the experimental results. Applying tissue engineering techniques might be helpful. The authors can tune the characteristics of the material properties, which can be controlled, and test the model under different conditions.
  3. Parameter Selection in Line 196:

    • In line 196, why are ωc and ω0 set to 16 and 60, respectively? The authors should explain the rationale behind choosing these specific values and how to select optimized parameters in the simulation process, as these choices significantly influence the results.
  4. Grammar and Language:

    • Grammar mistakes should be avoided. For example, in line 19, "visco-lastic" should be corrected to "viscoelastic." Polishing the language and grammar throughout the review will make it more professional.

By addressing these concerns, the paper will be more robust and provide clearer insights into the experimental design and its significance.

Comments on the Quality of English Language

 Moderate editing of English language required.

Round 2

Reviewer 3 Report

Comments and Suggestions for Authors

The authors have provided clear and comprehensive explanations and have adequately addressed all points of concern. Based on their responses and the revisions made to the manuscript, I recommend the paper in the Journal of Micromachines.